METHODS AND RESOURCES

# Stochastic models allow improved inference of microbiome interactions from time series data

**Román Zapién-Campos**[¤a]*, **Florence Bansept**[¤b], **Arne Traulsen**[*]

Max Planck Institute for Evolutionary Biology, Plön, Germany

¤a Current address: Centre for Life's Origins and Evolution, University College London, London, United Kingdom
¤b Current address: Aix Marseille Université, CNRS, Laboratoire de Chimie Bactérienne (UMR7283), IMM, IM2B, Turing Center for Living Systems, Marseille, France
* zapien@evolbio.mpg.de (RZ-C); traulsen@evolbio.mpg.de (AT)

**Data Availability Statement:** The authors confirm that all data underlying the findings are fully available without restriction. The data and software, including Jupyter Notebooks, used to generate the

## Abstract

How can we figure out how the different microbes interact within microbiomes? To combine theoretical models and experimental data, we often fit a deterministic model for the mean dynamics of a system to averaged data. However, in the averaging procedure a lot of information from the data is lost—and a deterministic model may be a poor representation of a stochastic reality. Here, we develop an inference method for microbiomes based on the idea that both the experiment and the model are stochastic. Starting from a stochastic model, we derive dynamical equations not only for the average, but also for higher statistical moments of the microbial abundances. We use these equations to infer distributions of the interaction parameters that best describe the biological experimental data—improving identifiability and precision. The inferred distributions allow us to make predictions but also to distinguish between fairly certain parameters and those for which the available experimental data does not give sufficient information. Compared to related approaches, we derive expressions that also work for the relative abundance of microbes, enabling us to use conventional metagenome data, and account for cases where not a single host, but only replicate hosts, can be tracked over time.

## Introduction

Numerous studies have shown how important the microbiome is for their hosts, ranging from development to health [1,2]. The promise of manipulating the microbiome relies on having understood the ecological and evolutionary processes operating on it [3]. Although metagenomics studies have widely characterized microbiome samples [4], their connection to mathematical models and eco-evolutionary theories lags behind. Part of the gap is explained by an intrinsic difficulty in analyzing microbiome data [5], in particular, the inverse problem of robustly inferring model parameters—and thus interactions between microbes—from data. Despite this difficulty, researchers have striven to enable the widespread use of parameter inference software in microbiome studies [6,7]. Pioneering work using linear regression to

results of this paper are available in Zenodo (https://doi.org/10.5281/zenodo.13958305). The mice microbiome data OMM12 are from https://doi.org/10.3389/fmicb.2019.02999 whose authors may be contacted at stecher@mvp.lmu.de.

**Funding:** We are grateful for funding from the German Science Foundation (Deutsche Forschungsgemeinschaft, DFG) within the Collaborative Research Center 1182 (Project-ID 261376515), project A4.1 (A.T.). The funders had no role in study design, data collection and analysis, decision to publish, or preparation of the manuscript.

**Competing interests:** The authors have declared that no competing interests exist.

infer interactions of the linearized Lotka–Volterra model [8] showed that matching the microbiome composition dynamics does not imply matching the true value of interactions in simulations [5]. This apparent contradiction stems from 2 challenges. First, in some models the value of individual parameters can not be told apart; this structural identifiability problem occurs even for infinite noiseless data [9]. Remien and colleagues showed, for example, that a Lotka–Volterra model of relative abundances is only locally identifiable; thus, without absolute abundance data, interactions can not be uniquely inferred in their deterministic model [10]. Second, as Cao and colleagues [5] discuss extensively, the fact that data is incomplete and the high dimensionality of the parameter space limit inference in practical ways. In addition, measurement noise of data makes the inference problem more challenging. There are indications that stochastic models, which track more statistical information than deterministic models, can overcome these challenges to some extent [11]. Using stochastic modeling, parameters were successfully inferred in systems biology [9,12] and cancer studies [13].

Here, we combine Bayesian inference—where probability distributions are inferred for the parameter values [14]—and stochastic modeling (akin to [9,12,13]) to improve parameter inference in microbiome studies. We propose a computational workflow that goes from microscopic transition rates in a mathematical model—describing ecological and evolutionary events (such as birth, migration, mutation, or speciation)—to macroscopic dynamics of the statistical moments of microbiome composition [9,12,13], see Fig 1. This Bayesian inference workflow, which naturally bypasses known limitations of linear regression (i.e., point value) inference [5], is sufficiently flexible to test different mathematical models and microbiome samples while quantifying the parameter uncertainty stemming from data limitations (Fig 1C), including measurement noise. We use 2 classical ecological models—logistic growth and the Lotka–Volterra model—to illustrate its application on data sets describing absolute or relative abundances of microbes. For the relative abundance models, we show that our workflow overcomes non-identifiability of communities with a small number of types, enabling parameter inference from conventional metagenome data. The workflow outlined here bridges a gap between microbiome data and theoretical modeling by addressing fundamental and practical aspects to infer microbial interactions.

## Results

### Developing an inference workflow

We propose a parameter inference workflow grounded on a mechanistic description of the dynamics of absolute abundances in a microbiome (Fig 1C). For simplicity, let us define a vector $\mathbf{n}$, where each element corresponds to the population of a microbial type. We can write down microscopic transition rates $T$ describing changes in the microbiome composition of one host, $\mathbf{n}$, to other compositions $\mathbf{n}'$, given the set of parameters $\boldsymbol{\theta}$,

$$T(\mathbf{n} \rightarrow \mathbf{n}') = f(\mathbf{n}, \boldsymbol{\theta}). \tag{1}$$

Now, instead of tracking the microbiome composition $\mathbf{n}$ in a single host, we can describe how the probability of a microbiome composition $\mathbf{n}$ in an ensemble of hosts $P(\mathbf{n}, t)$, changes with time,

$$\frac{\partial P(\mathbf{n}, t)}{\partial t} = \underbrace{\sum_{\mathbf{n}'} P(\mathbf{n}', t) T(\mathbf{n}' \rightarrow \mathbf{n})}_{\text{probability influx to } \mathbf{n}} - \underbrace{\sum_{\mathbf{n}'} P(\mathbf{n}, t) T(\mathbf{n} \rightarrow \mathbf{n}')}_{\text{probability outflux from } \mathbf{n}}. \tag{2}$$

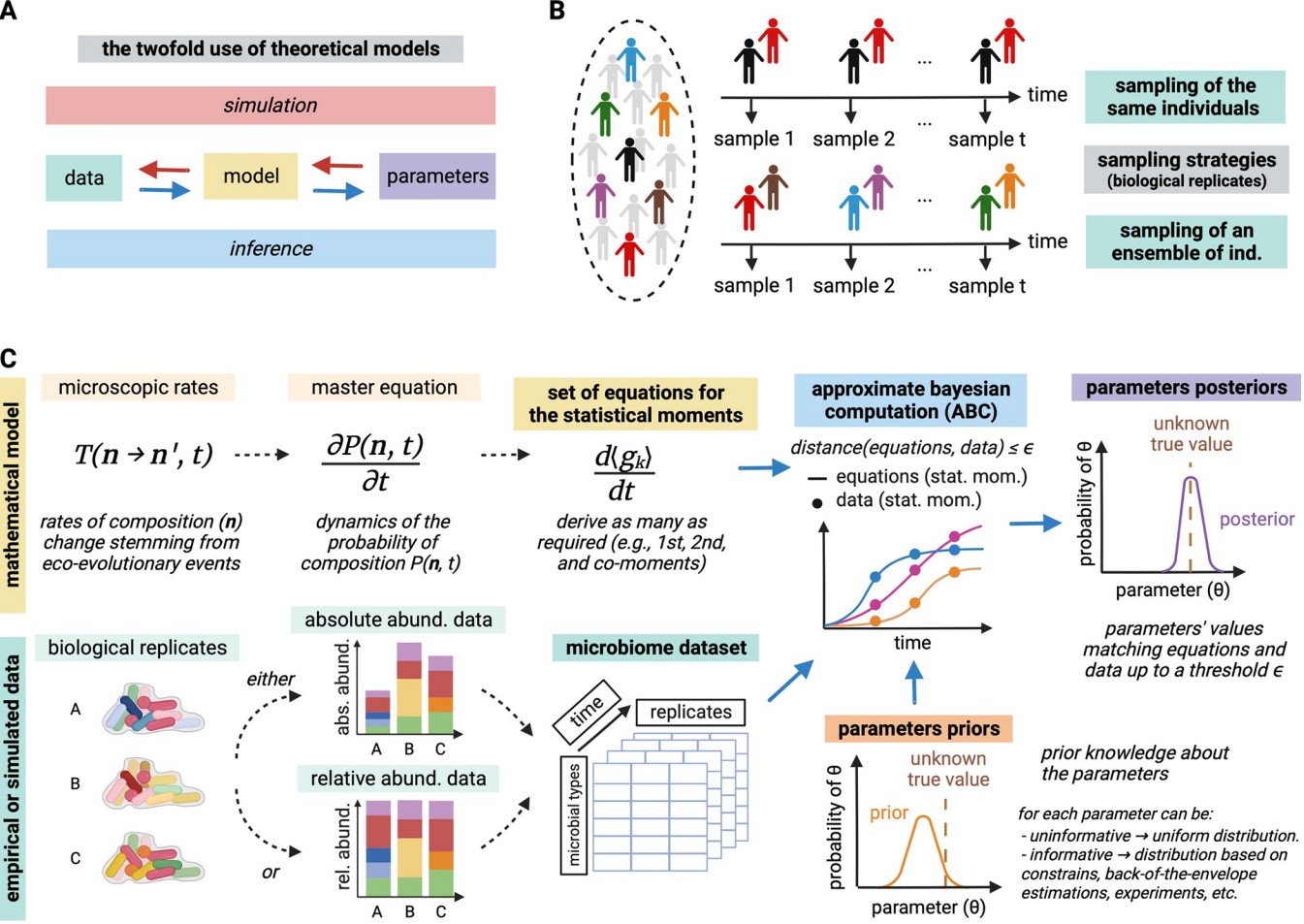

**Fig 1. Parameter inference workflow proposed and microbiome data properties.** (**A**) Mathematical models serve as a link between parameters and data. Either to simulate biological processes or to infer parameters from data. (**B**) Longitudinal sampling of the same hosts or an ensemble of them are used to obtain datasets. (**C**) Workflow from microscopic rates of a model and experimental data to inference of parameters values by ABC. The microscopic rates describe possible eco-evolutionary events (such as birth, migration, mutation, or speciation), leading to macroscopic patterns (statistical moments of abundance). Data sets describe absolute abundances (counts) or relative abundances (frequencies) of microbes. To quantify the probability of parameter values given a data set, prior knowledge about the parameters is updated to a posterior distribution based on the agreement of the model with the data. Note that because the model describes the dynamics continuously, no correlation between time points is needed. Figure created in BioRender.com under a CC-BY-NC-ND license.

This expression, called the master equation [15], allows us to compute statistical information about the microbiome composition beyond its mean behavior. Here, the probability influx and outflux terms indicate an increase or decrease in the probability of composition **n** caused by transitions from and to other microbiome compositions (**n′**). Therefore, the dynamics of the microbiome depend on the ecological and evolutionary processes contained in the transition rates.

Using the master equation, we derive equations for the statistical moments of the microbiome composition in an ensemble of hosts—namely, the product of the master equation by a variable of interest ($g_k$, where $k$ is an identifying index) summed over all possible microbiome compositions **n**,

$$\frac{d\langle g_k\rangle}{dt} = \sum_{\mathbf{n}} g_k \frac{\partial P(\mathbf{n}, t)}{\partial t}.$$

(3)

This is a way to average a variable from the model. For example, computing the average abundance of microbial type $i$ implies setting $g_k = n_i$. If we set $g_k = n_i n_j$, we obtain an equation for the co-moment of microbial types $i$ and $j$. The resulting equations describe the expected macroscopic dynamics of the microbiome: tracking a large stochastic system without an explosive computational burden. Now, to extract sufficient statistical information from the model we can derive several of these equations, even as many as the number of free parameters. For example, in a Lotka–Volterra model with $S$ microbial types, there are $S$ growth rates and $S^2$ intra- and inter-specific interactions, amounting to $S + S^2$ parameters. We could derive $S + S^2$ equations to match the number of parameters, including, $S$ equations for the first moments $\langle n_k \rangle$, $S$ for the second moments $\langle n_k^2 \rangle$, and $S(S-1)$ for the co-moments $\langle n_k n_l \rangle$ and covariances $\langle n_k, n_l \rangle$ (see the Methods). Note that each equation can depend on the vector of other moments, i.e., $\langle g_k \rangle = f(\langle \mathbf{g} \rangle, \boldsymbol{\theta}, t)$. While in some models moments will depend on moments of equal or lower order ("closed equations"), in others they will depend on even higher order moments, leading to an infinite system of interdependent equations. Because closure is required to solve any system of equations, we illustrate how to approximate higher-order moments in the Methods. Note that in spite of the large system of equations to solve, our approach exploits the fact that, except from "closed equations," many equations are linear thus quickly solved by conventional ODE solvers. Here, we presented only the generic derivation of the workflow; a step-by-step derivation from microscopic rates up to second-order moments for a logistic growth and the Lotka–Volterra models can be found in the Methods. Such models include conventional ecological events, such as growth, death, immigration, and direct and indirect interactions.

We now have the elements to infer the parameters $\boldsymbol{\theta}$ from microbiome data. The focus now switches to the fitting method, with 2 possibilities: likelihood-based methods such as Markov Chain Monte Carlo (MCMC) [16] or likelihood-free methods such as Approximate Bayesian Computation (ABC) [17] which use the dynamical equations instead. Here, we opt for ABC as true likelihoods of stochastic models can rarely be derived [14]; however, MCMC assuming a pseudo-likelihood (e.g., a Gaussian likelihood) can be a promising alternative to optimize computational efficiency. The idea of ABC is to identify feasible parameters values by comparing the data to dynamical model predictions [14]. Specifically, for any given set of parameters values $\boldsymbol{\theta}$, a distance metric between the numerical solution of the equations for the moments, $\langle g_k \rangle$, and the equivalent moments from data, $\bar{g}_k$, is estimated, e.g.,

$$\sum_k \sum_i | \underbrace{\langle g_k \rangle (\langle \mathbf{g} \rangle, \boldsymbol{\theta}, t_i)}_{\text{model solution}} - \underbrace{\bar{g}_k(t_i)}_{\text{data}} |, \tag{4}$$

for the Euclidean distance (the effect of rescaling some moments is shown in S1 Fig), where the sum over $i$ refers to the data points, and the sum over $k$ refers to the different moments. If this distance is smaller than a threshold $\varepsilon$, the set $\boldsymbol{\theta}$ is considered to be a valid parameter estimate. By testing sets of parameters sampled according to an expectation—the prior distribution—and recording those below the threshold $\varepsilon$, a posterior distribution of the parameters reflecting the uncertainty of the inference can be obtained (Fig 1C). With a smaller threshold $\varepsilon$, this posterior can become the new prior and the process can be iterated to narrow down the parameter distributions. This method is called Approximate Bayesian Computation—Sequential Monte Carlo (ABC-SMC). We show how to choose prior distributions of the parameters in Tables 3–5.

## Properties of microbiome data

Given a microbiome data set of abundances with replicates, all statistical moments $\bar{g}_k$ can be estimated from it. Concretely, this is done by averaging the variable of interest $g_k$, over all

replicates in each specific time point (Fig 1C). For example, for $g_k = n_i$ the replicates of $n_i$ are summed over and divided by the number of replicates, while for $g_k = n_i n_j$, the products of $n_i$ and $n_j$ for each replicate are computed, then summed over and divided by the number of replicates.

Microbiome data is nowadays typically produced by metagenome sequencing. Conventionally, for technical reasons, metagenomics only quantifies the relative abundance of each microbial type in a sample (Fig 1C) [18]. More recently, some studies have measured absolute numbers of culturable [19] and non-culturable microbes in samples [20]. We call these counts absolute abundances.

Our former equations only track moments of absolute abundance, $\langle g_k \rangle$. As Gloor and colleagues [18] show, inferring parameters from relative abundance ($x_k$) data using these would lead to spurious correlations (Fig 2A and 2B). To find equivalent expressions for the statistical moments of relative abundance, we define $n_\Sigma \equiv \Sigma_j n_j$, the total microbiome population, and the dynamical equation for its first moment, $\langle n_\Sigma \rangle$, to be used as a scaling factor. A transformation to moments of relative abundances, $\langle \gamma_k \rangle$, is given by

$$\frac{d\langle \gamma_k \rangle}{dt} = f(\langle \boldsymbol{\gamma} \rangle, \langle n_\Sigma \rangle, \boldsymbol{\theta}, t). \tag{5}$$

Because relative abundance data sets lack information about the scaling factor, its initial condition, $\bar{n}_\Sigma(0)$, must be inferred as a free parameter, one parameter more than for absolute abundance data. This scaling factor can be the quantity of interest sometimes [21]. Note that because the relative abundances add up to one, $\Sigma_k x_k = 1$, the number of independent equations for the microbial types decreases by 1, but the number of parameters per type remains. A detailed derivation of transformations to relative abundance for a logistic growth and the Lotka–Volterra models is shown in the Methods.

While some studies track the microbiome of the same host over time, in many microbiome studies, replicate hosts are sampled at different time points and pulled together to produce a single time series (Fig 1B). This is the case when hosts are sacrificed while sampling as in experimental studies of *Drosophila melanogaster*, *Caenorhabditis elegans* [22], and *Hydra vulgaris* [19,23]. In contrast to deterministic models, the workflow shown here can deal with hosts pulled together as it accounts for stochastic demography. Concretely—akin to the concept of biological replicates—if the parameter values and initial conditions are the same in each host sampled, we can account for their emerging demographic differences, i.e., expected differences in microbiome composition resulting from a stochastic reality.

Finally, our workflow does not make assumptions about the experimental technology to obtain microbial abundance data. However, it is important to be aware of potential biases introduced while obtaining and preprocessing raw data [24].

## The advantages of our workflow for inference

Deriving dynamical equations for the moments is more cumbersome than writing down deterministic equations. Nevertheless, the additional effort pays back on inference in at least 2 ways:

1. Firstly, the dynamics of the moments use more information contained in the data, increasing the chance of estimating the true parameter values (Fig 3).

2. Secondly, the larger number of equations and their structural differences can improve the structural identifiability of the parameters, guaranteeing that for infinite noiseless data their unique value can be known.

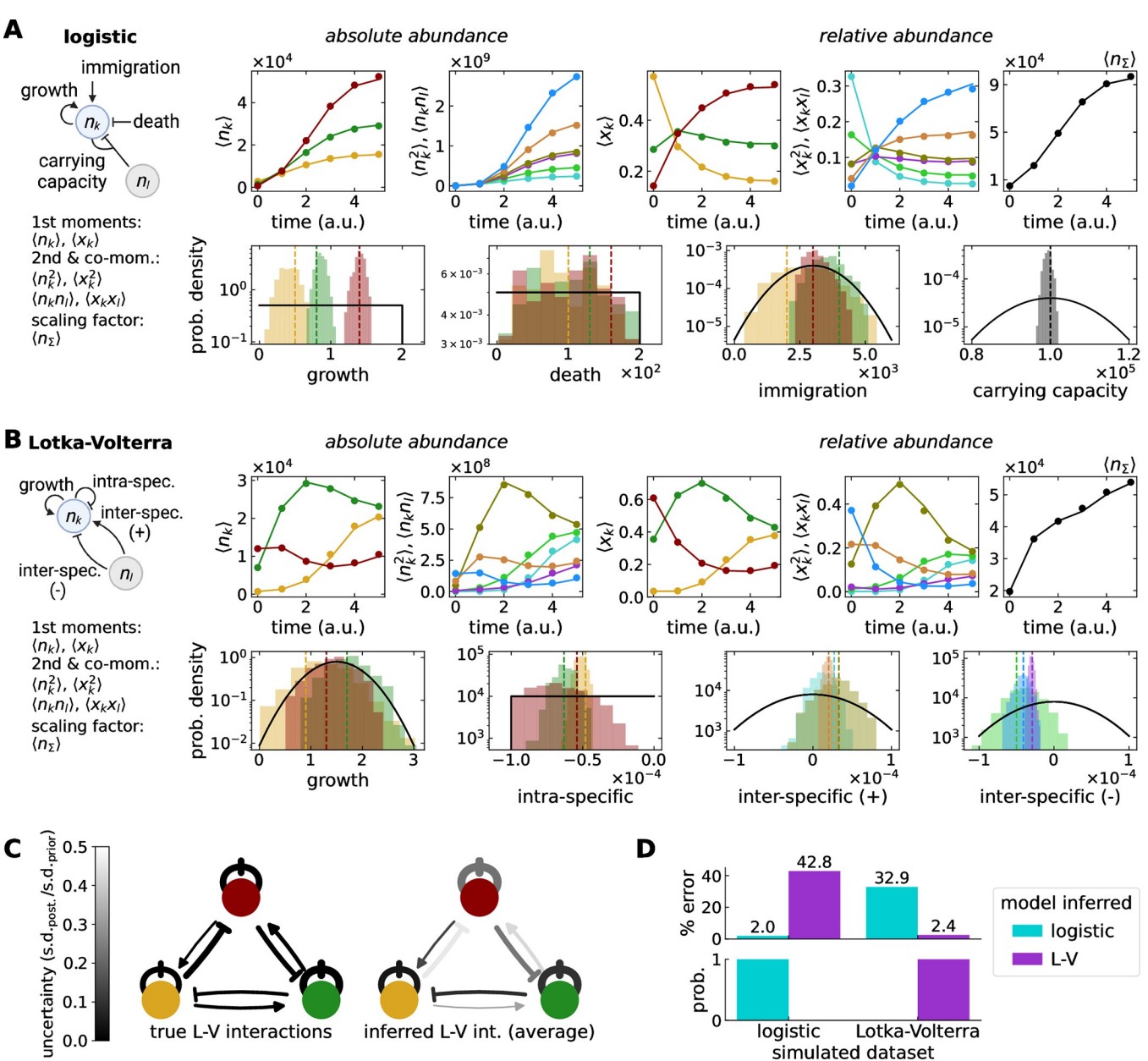

**Fig 2. Inferring true parameters from simulated data.** (**A, B**) Time series comparison between simulations (dots, derived from only 4 replicates) and equations for the statistical moments (lines) of absolute ($n_k$) and relative abundance ($x_k$) sharing true parameters (found in Tables 1 and 2). Two models with 3 microbial types ($S = 3$) were tested, (**A**) logistic growth with immigration and death $3S + 1 = 9$ parameters and (**B**) Lotka–Volterra $S + S^2 = 12$ parameters. Inferred parameter posteriors from the relative abundance are compared to true parameters (dashed lines) and priors (black distributions). All microbial types shared the same priors (Tables 3 and 4). (**C**) The inferred interactions for the Lotka–Volterra model resembled the true interactions, qualitatively (arrowheads) and quantitatively (arrow thickness), with various certainties (grayscale, defined by the ratio of SD of posterior to prior). (**D**) For both data sets, the most probable model was identified correctly. The settings for the inference are listed in Table 6 (a.u. = time units are determined by the rates, see Tables 1 and 2). Networks on the left of **A** and **B** were created in BioRender.com under a CC-BY-NC-ND license. The data underlying this figure can be found in https://doi.org/10.5281/zenodo.13958305.

Using identifiability software, Browning and colleagues [9] showed that parameters can turn identifiable when dynamical moments are considered. Such gain depends on the combined effect of the number of equations, sampled time points, and latent (non-measured) variables. We used GenSSI [25], a Matlab package that uses series and tableaus to test the

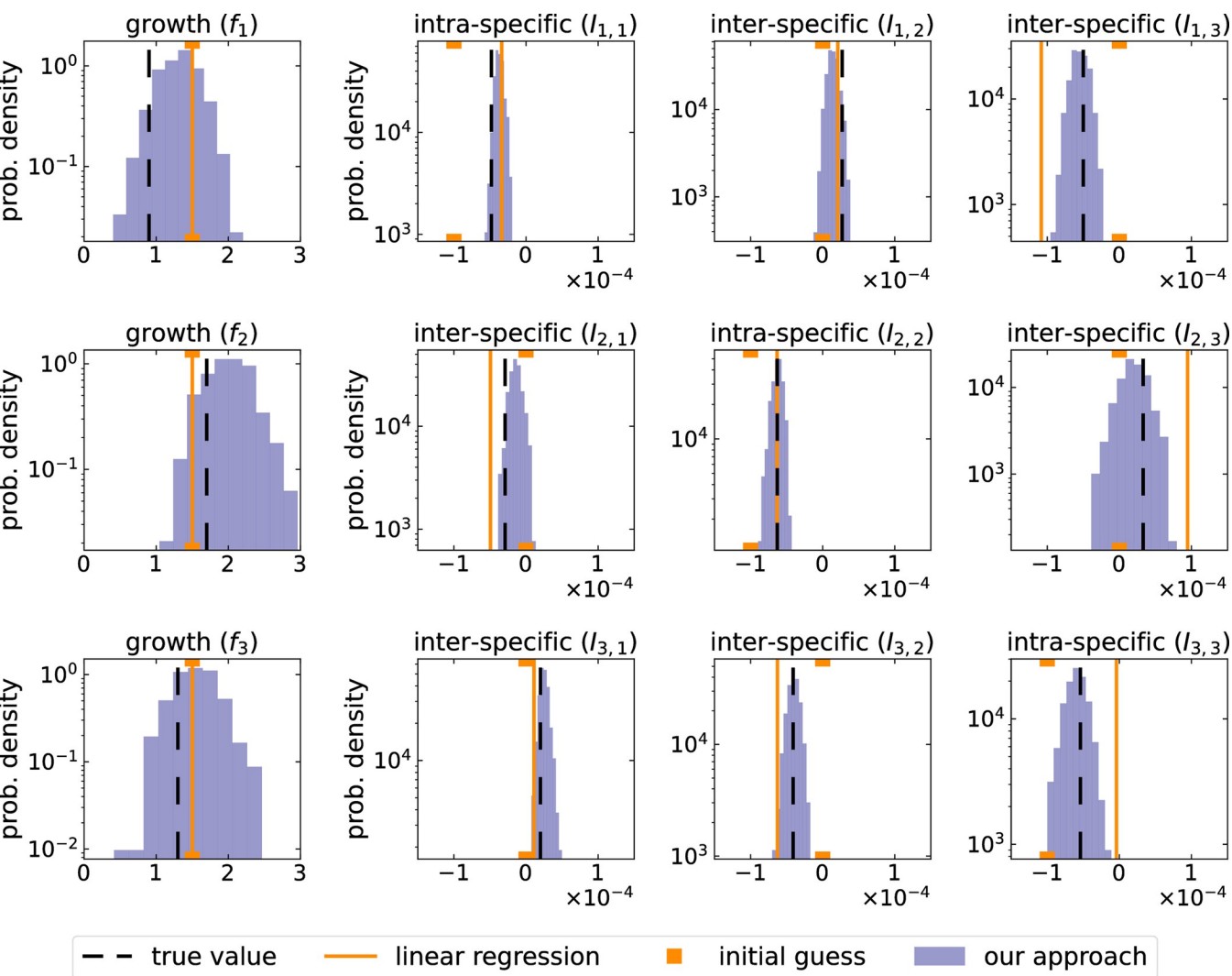

**Fig 3. Outcome comparison of our workflow and linear regression of the deterministic Lotka–Volterra model.** We inferred all parameter values (Table 2) from simulated absolute abundance data as in Fig 2. While our workflow used the same setup of Fig 2, the linear regression method was based on [8] without time-dependent perturbations or regularization. Our Bayesian workflow successfully "locates" the true parameter values, along their uncertainty, even if the linear regression method does not. The initial parameters guess for linear regression was close to the true value (1.5 for growth rates, and $-10^{-4}$ and 0 for intra- and inter-specific interactions). For our workflow, we used the same parameter priors of Fig 2, summarized in Table 4. The data underlying this figure can be found in https://doi.org/10.5281/zenodo.13958305.

identifiability of a model (Fig 2 and Methods). Its expansion of the dynamical model around sampled points to extract the information available of the parameters is one of the most used methods for nonlinear systems [26,27]. We found that for absolute abundance, Lotka–Volterra is globally identifiable, while logistic growth has finite possible values, thus, locally identifiable. The relative abundance models retained these identifiability categories, improving the local identifiability reported for a deterministic Lotka–Volterra model [10].

Overall, statistical moments can improve structural identifiability [9], narrowing down the success of inference to the properties—quality and amount—of the data. In the following, we illustrate this practical aspect with guarantees of improved identifiability and inference of parameters from absolute and relative abundance microbiome data.

## Inference from simulated and empirical data

We tested our inference workflow in 2 ways. Firstly, we inferred parameters from simulated relative abundance data (Fig 2) to compare each inferred value to their true known value. Our approach, with three microbial types, proved successful in models with and without inter-specific interactions, namely, data from Lotka–Volterra and logistic growth simulations. In fact, in contrast to linear regression using a deterministic model [8] our approach "located" the true Lotka–Volterra interaction values every time (Fig 3). The uncertainty of the parameters reflected the limitations of the data, e.g., death rates being more uncertain as a result of data only tracking a growth phase (Fig 2A). Beyond parameter values and certainty, we were able to identify the correct data-generating model each time. Importantly, only 6 time points and only 4 replicates were included in each data set, a realistic scenario for experimental studies.

Measurement noise can increase the uncertainty of inferred values. To test our workflow, we inferred parameters from simulated Lotka–Volterra data with increasing amounts of noise (Figs 4 and S2). Although noise can be influenced by many factors [24], we focused on a case where a shared noise distribution affects each microbial type at each time point. Inferring parameters from relative abundance data led to larger uncertainty than inferring from absolute abundance data. However, in both cases, uncertainty was reduced by having more replicates and/or time points (Fig 4), with the number of time points having a stronger effect.

The encouraging results from simulations led us to apply our inference workflow to experimental data. We used for this a thoroughly measured time series of replicates with a small number of microbial types: the absolute abundance data of OMM$^{12}$—a reduced mice microbiome [28] (Fig 5). Such data set tracks the growth of microbes in the gut from a germ-free state. We used the logistic growth model, which describes transient and equilibrium stages, to illustrate our approach. The inferred posteriors suggested the growth rates of *Akkermansia muciniphila*, *Bacteroides caecimuris*, *Bifidobacterium longum*, and *Muribaculum intestinale* to be most certain, with average doubling times ranging from hours to days. Meanwhile, except from *B. caecimuris*, the average death and immigration rates were less certain, ranging from $\approx 4 \cdot 10^5$ to $1.4 \cdot 10^6$ cells per day. Most of the certainties obtained from empirical data (Fig 5) are smaller than those from simulations (Fig 2), highlighting the limits of the model tested and inference from noisy, empirical data. However, in each case, we obtained a set of parameters—capturing interactions between microbes—with some level of certainty. Our results point to selection as the ecological driver of the OMM$^{12}$ dynamics, despite a possible compatibility of this data with a neutral hypothesis once it has reached steady state [29]. In this case, neutrality would imply that the parameter posteriors overlap between all microbial types, which is not the case (Fig 5).

## Discussion

Our work is motivated by the goal of understanding how microbes interact and the need to quantify the uncertainty of parameters (interactions) inferred from microbiome data. Although point-value inference methods have been used previously [8], several issues limit their quantitative application, restricting them to recreate qualitative patterns of data [5]. A major issue is that models often have more parameters than equations [5,8,30], limiting the information to constrain the large number of interactions to infer. We propose a solution by deriving equations for the statistical moments of the microbiome composition—even as many as the number of parameters—to make better use of the information contained in the data. Supporting this idea, statistical moments have improved inference from molecular [12] and cancer data [13]. Browning and colleagues [9] found that statistical moments improve the structural identifiability of parameters in simulations, which we confirmed for logistic growth

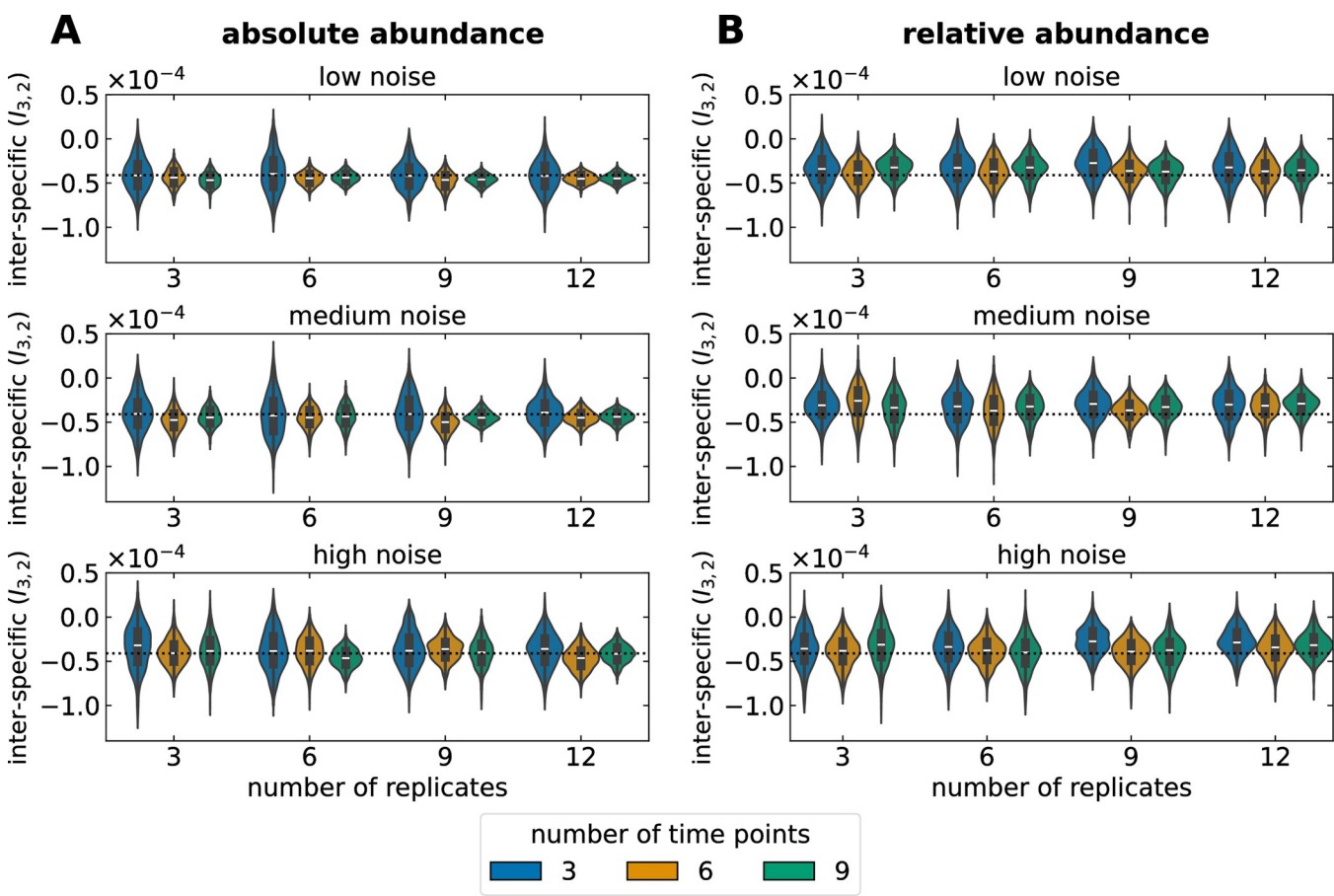

**Fig 4. Effect of data measurement noise on the uncertainty of an inferred Lotka–Volterra parameter.** We inferred all parameters from simulated data as shown in Fig 2 (see Table 2). For simplicity, we show the effect of noise on a single parameter with true value $I_{3,2} = -4.1 \cdot 10^{-5}$ (dashed line). The effect on all parameters is shown in S2 Fig. We simplified the nuances of empirical noise [24] assuming a scenario where all microbial abundances are affected proportionally. Concretely, a uniform noise distribution was shared among all microbial types and constant through time. For low noise, data could be altered by up to ±5%, while for medium and high noise, by up to ±10% and ±20%. Noise was sampled independently for each microbial type at each time point, affecting the absolute abundances from which relative abundances were computed. (**A**) A larger number of replicates and/or time points help reduce the increased uncertainty caused by noise. In particular, the number of time points has a stronger effect than the number of replicates. (**B**) The uncertainty obtained from relative abundance data is consistently larger than from absolute abundance. Still, more replicates and/or sampling time points help to reduce the uncertainty. The data underlying this figure can be found in https://doi.org/10.5281/zenodo.13958305.

and Lotka–Volterra models of absolute and relative abundance [10]. Our approach is driven by a mechanistic spirit, where microscopic rates must be written down first, based on hypothetical mechanisms and stating assumptions. As opposed to approaches where analytic solutions—or expensive stochastic simulations—are needed, here, a numerical solution is sufficient to quantify the distance between equations and data, despite the large number of parameters, microbial types, and population size [17]. This allows our workflow to handle diverse models, where formal model comparison is possible [12,17].

The workflow is not limited by the properties of the microbiome abundance data [18]. As we have shown, analyzing data sets describing the relative abundance of microbial types—even if the total absolute abundance is dynamic [30]—is possible. Such is the nature of metagenomic sequencing data—the most common method to characterize microbiomes [5]. In addition, by tracking statistical moments of the microbiome, our approach naturally accounts for the diverse types of experimental samplings, such as those where ensembles of hosts are used to

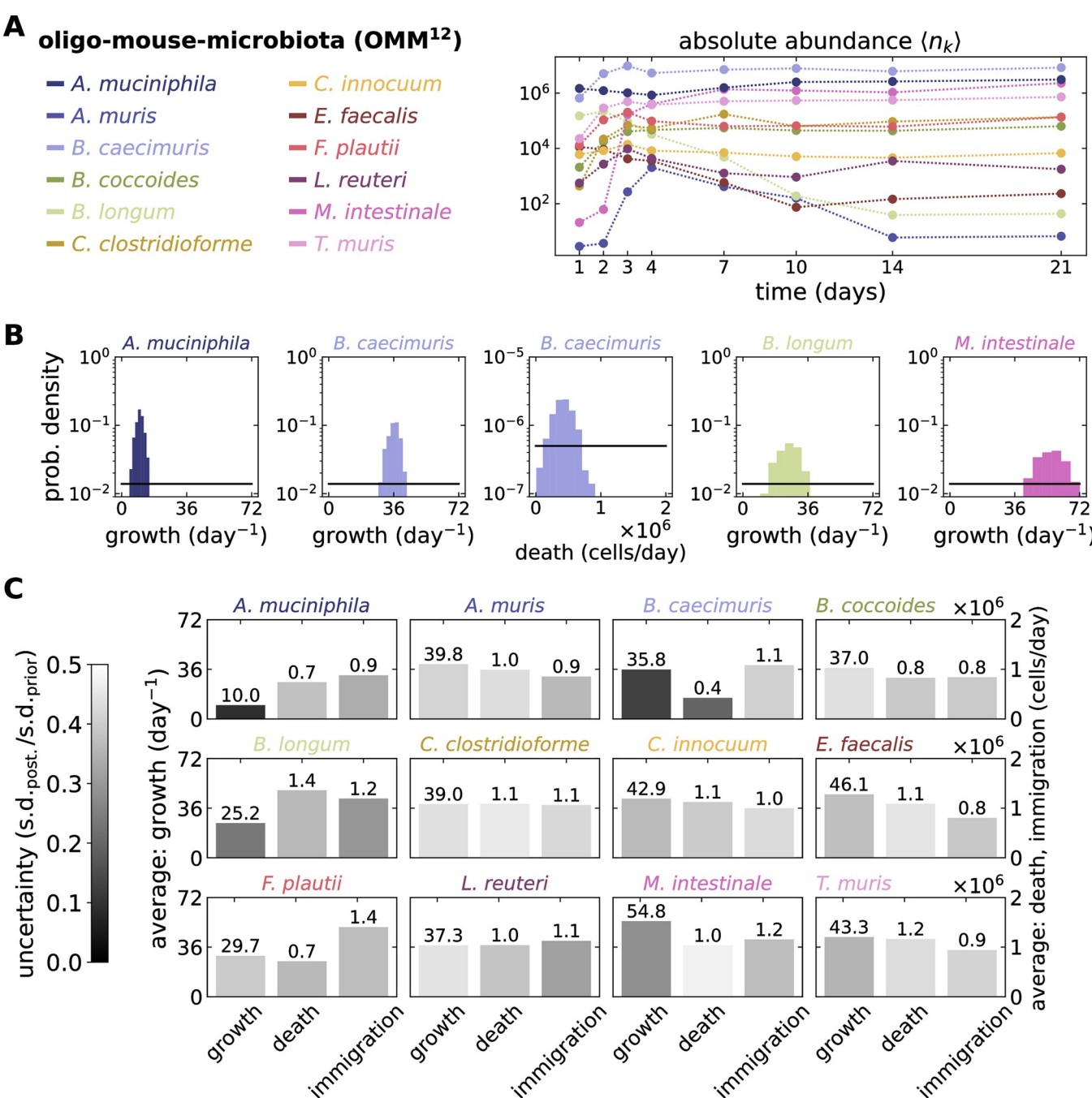

**Fig 5. Inferring parameters from empirical data.** The parameters of a logistic growth with immigration and death model were inferred from a mouse dataset. The Oligo-Mouse-Microbiota (OMM[12]) data set [28] tracks a 12-species defined mice microbiome ($S = 12$), where the absolute abundances in the same individuals were sampled from feces 11 times over 99 days. (**A**) We analyzed the first 21 days where 4 replicates are available, we show here the abundance of all 12 types averaged over the 4 replicates. We use the underlying data to infer the parameters of a logistic growth model with growth, death, and immigration, with in total $S + S^2 = 156$ moments used for the inference. (**B**) Of the $3S + 1 = 37$ parameters inferred, we show only the posteriors of the 5 most certain ones (defined by the ratio of SD of posterior to prior as a relative comparison of the certainty gained between parameters). All microbial types shared the same uniform priors (black lines, Table 5) to have a fair measure of the parameter uncertainty reduced. (**C**) The parameters inferred for each species varied widely with various certainties. For the shared carrying capacity, we found an average $N \approx 1.45 \cdot 10^7$ bacterial cells, $\pm 3.49 \cdot 10^5$ cells, and uncertainty of 0.0582. A system of 156 equations was solved ($S = 12$ first moments and $S^2 = 144$ second moments and co-moments). The settings for the inference are listed in Table 6. The data underlying this figure can be found in https://doi.org/10.5281/zenodo.13958305.

obtain a single time series. Concretely, compared to other methods, we track the demographic variation between hosts explicitly and assign the remaining variation to external "environmental" noise. Measurement noise can be incorporated using knowledge about its distribution, including: shape, dynamics, and how each microbial type is affected [9,10,13]. Missing this information, we did not consider a noise model for the OMM[12] data set. However, we showed that parameter uncertainty from simulated noisy data could be reduced by increasing the number of time points first, or the number of replicates second. Still, other considerations including the interval between time points and capturing the transient dynamics could be important to overcome the effect of noise in empirical data. For example, having more replicates might be more beneficial for nearly steady-state dynamics.

Our workflow assumes that samples originate from the same environmental and initial conditions. To date, such replicates are more easily obtained in laboratory conditions [28]. However, antibiotics, microbiome transplantations, or other perturbation treatments could be explored as means to force the generation of replicates. Alternatively, models of higher taxonomic levels, where microbial compositions are more similar [24], could be written.

Although by design, our workflow deals with common longitudinal (time series)—even sparse—data, analyzing a single time point (snapshot) is in principle possible. For example, if the microbiome composition is assumed to be at a steady state, the inference method's aim is to find parameters making the dynamical equations for the moments equal to zero. This does not mean that the moments are zero, but that their rate of change is. This differs from quasi-steady data, which is common in microbiome studies [31,32], but less informative than non-steady data. However, single time points are not expected to be as effective as multiple time points. As our results illustrate, given the various sources of uncertainty, non-steady data leads to better parameter inference, in particular, those time intervals of "high activity" where many compositional changes occur [5]. As Cao and colleagues [5] proposed, several of these intervals could be analyzed simultaneously to improve the inference.

Bayesian inference can suffer the curse-of-dimensionality in large and diverse systems [17]. By combining statistical moments readily solved numerically and data of sufficient quality, we believe our workflow can overcome this to some extent, exploring the parameter space in a feasible time. Physical and biological constraints can focus the parameter exploration further [33]. We implemented an Approximate Bayesian Computation with Sequential Monte Carlo in our workflow using tools from the Python package pyABC [34] (Table 6). Other optimizations, or combinations with methods such as Markov Chain Monte Carlo [16], could greatly improve its wider application [14]. As proof-of-principle, we applied our workflow to 2 simulated relative abundance datasets and recovered the true parameter values. We also applied it to a reduced microbiome in mice [28], where we estimated values and certainties of parameters describing logistic growth, a quantitative characterization of the microbes in situ. Our contribution builds towards the aim of developing tools to enable the widespread use of parameter inference in microbiome studies, where large progress has been made (see, e.g., [6,7]).

Although Lotka–Volterra and logistic growth are meaningful ecological models to investigate first, other alternatives can be tested as well. For example, a model of logistic growth with linear environmental noise, different from our demographic-noise-only model, suggests that environmental perturbations determine many properties of the microbiome composition [31,32]. Despite only considering time-independent pairwise interactions, our workflow can incorporate higher-order interactions [35], as well as time-dependent [36] or time-delayed interactions [37] in the transition rates. Even multilayer networks [38], i.e., assorted interactions, can be modeled as we illustrate separating positive and negative interactions in a Lotka–Volterra model. Similarly to the model comparison between Lotka–Volterra and logistic

growth in our results, contrasting alternative models could point to the underlying mechanisms operating in microbiomes.

In summary, we presented a Bayesian inference workflow bridging microbiome data to theoretical modeling. We used the ability of stochastic models to track statistical quantities beyond mean behaviors [9,11–13], enabling us to exploit useful information contained in dynamical data. This workflow can be facilitated by existing automated software to derive statistical moments from dynamical models [39]. By inferring from data sets of microbial absolute and relative abundances, we showed its robustness—identifying likely interactions and certainty of parameters in simulated and empirical data. Because mechanistic rates serve as stepping stones of the workflow, similar microscopic models could replace the 2 classical ecological models that we illustrated—including experimentally informed models.

## Methods

### Derivation of dynamical equations for the microbiome moments

To track the statistical moments of a model, e.g., average, variances, and co-variances, we have to account for the stochasticity of events. Thus, describing the probability of microbiome compositions is needed. The change in probability of each microbiome composition is described by the master equation,

$$
\frac{\partial P(\mathbf{n}, t)}{\partial t} = \underbrace{\sum_i P(\mathbf{n} + \mathbf{e}_i, t) T(\mathbf{n} + \mathbf{e}_i \to \mathbf{n})}_{\text{influx from states } n_i+1} + \underbrace{\sum_i P(\mathbf{n} - \mathbf{e}_i, t) T(\mathbf{n} - \mathbf{e}_i \to \mathbf{n})}_{\text{influx from states } n_i-1} \\
- \underbrace{\sum_i P(\mathbf{n}, t) T(\mathbf{n} \to \mathbf{n} + \mathbf{e}_i)}_{\text{outflux to states } n_i+1} - \underbrace{\sum_i P(\mathbf{n}, t) T(\mathbf{n} \to \mathbf{n} - \mathbf{e}_i)}_{\text{outflux to states } n_i-1},
$$

(6)

where $\mathbf{n}$ is the vector of absolute microbial abundances, and $\mathbf{e}_i$ is the amount of change, a vector with one in the *i-th* entry and zero otherwise.

Dynamical equations for the statistical moments can be obtained from the master equation by multiplication and subsequent summation; e.g., for the first moment $\langle n_k \rangle$, equivalent to the average, we have

$$
\frac{d\langle n_k \rangle}{dt} = \sum_{\mathbf{n}} n_k \frac{\partial P(\mathbf{n}, t)}{\partial t} = \dots \sum_{n_k=0}^{\infty} \dots n_k \frac{\partial P(\mathbf{n}, t)}{\partial t},
$$

(7)

where for clarity, we make summations more explicit. For the second moment $\langle n_k^2 \rangle$, we have

$$
\frac{d\langle n_k^2 \rangle}{dt} = \sum_{\mathbf{n}} n_k^2 \frac{\partial P(\mathbf{n}, t)}{\partial t} = \dots \sum_{n_k=0}^{\infty} \dots n_k^2 \frac{\partial P(\mathbf{n}, t)}{\partial t},
$$

(8)

and for the co-moments $\langle n_k n_l \rangle$,

$$
\frac{d\langle n_k n_l \rangle}{dt} = \sum_{\mathbf{n}} n_k n_l \frac{\partial P(\mathbf{n}, t)}{\partial t} = \dots \sum_{n_k=0}^{\infty} \sum_{n_l=0}^{\infty} \dots n_k n_l \frac{\partial P(\mathbf{n}, t)}{\partial t}.
$$

(9)

For models with a finite carrying capacity, the upper sum limit is changed to a finite number.

## Logistic growth with immigration and death

Let us exemplify the former steps with a logistic growth model. Similarly to Allouche and colleagues [40], let us define the microscopic transition rates for one microbial population $i$,

$$T(\mathbf{n} \rightarrow \mathbf{n} + \mathbf{e}_i) = (f_i n_i + m_i) \frac{N - \sum_j n_j}{N} \tag{10a}$$

$$T(\mathbf{n} \rightarrow \mathbf{n} - \mathbf{e}_i) = \phi_i \frac{n_i}{N}, \tag{10b}$$

where $N$ is the maximum number of microbes in a host (shared carrying capacity), $f_i$ is the maximum growth rate, and $\phi_i$ and $m_i$ are the death and immigration rates for each type $i$. We assume small death rates $\phi_i$ (Table 1), following the typical logistic growth concept, where only birth occurs. But, in addition, close to $\sum_j n_j \approx N$ death occurs. In such limit, the model resembles a death-birth process where the microbial abundances ($\mathbf{n}$) slowly move towards an equilibrium less influenced by the initial abundances but more by the rates of birth, death, and immigration [41].

Now, we illustrate how to derive dynamical equations for the moments. Let us start with the first moment,

$$
\begin{aligned}
\frac{d\langle n_k \rangle}{dt} = \quad & \ldots \sum_{n_k=0}^{N-1} \ldots n_k P(\mathbf{n} + \mathbf{e}_k, t) T(\mathbf{n} + \mathbf{e}_k \rightarrow \mathbf{n}) \\
& + \ldots \sum_{n_k=1}^{N} \ldots n_k P(\mathbf{n} - \mathbf{e}_k, t) T(\mathbf{n} - \mathbf{e}_k \rightarrow \mathbf{n}) \\
& - \ldots \sum_{n_k=0}^{N} \ldots n_k P(\mathbf{n}, t) T(\mathbf{n} \rightarrow \mathbf{n} + \mathbf{e}_k) \\
& - \ldots \sum_{n_k=0}^{N} \ldots n_k P(\mathbf{n}, t) T(\mathbf{n} \rightarrow \mathbf{n} - \mathbf{e}_k) \\
& + \ldots \sum_{n_k=0}^{N}\sum_{n_i=0}^{N-1} \ldots n_k \sum_{i \neq k} P(\mathbf{n} + \mathbf{e}_i, t) T(\mathbf{n} + \mathbf{e}_i \rightarrow \mathbf{n}) \\
& + \ldots \sum_{n_k=0}^{N}\sum_{n_i=1}^{N} \ldots n_k \sum_{i \neq k} P(\mathbf{n} - \mathbf{e}_i, t) T(\mathbf{n} - \mathbf{e}_i \rightarrow \mathbf{n}) \\
& - \ldots \sum_{n_k=0}^{N}\sum_{n_i=0}^{N} \ldots n_k \sum_{i \neq k} P(\mathbf{n}, t) T(\mathbf{n} \rightarrow \mathbf{n} + \mathbf{e}_i) \\
& - \ldots \sum_{n_k=0}^{N}\sum_{n_i=0}^{N} \ldots n_k \sum_{i \neq k} P(\mathbf{n}, t) T(\mathbf{n} \rightarrow \mathbf{n} - \mathbf{e}_i),
\end{aligned}
\tag{11}
$$

**Table 1. Parameters in simulated logistic growth with immigration and death (Fig 2A).** The growth and death rates as well as the immigration parameters were only chosen for illustration, thus, time units are arbitrary. We used a relatively small population size for simplicity. However, larger population sizes can be easily tested.

| Microbial type | Growth $f_i$ | Death $\phi_i$ | Immigration $m_i$ | Shared carrying cap. $N$ | Initial pop. size |
|---|---|---|---|---|---|
| $i = 1$ | 0.5 | 100 | 2,000 | $10^5$ | 2,800 |
| $i = 2$ | 0.8 | 130 | 4,000 | $10^5$ | 1,400 |
| $i = 3$ | 1.4 | 160 | 3,000 | $10^5$ | 700 |

where the first 4 lines describe birth or death of a microbe of type $k$ and the last 4 lines describe birth or death of a microbe of type $i$ different from $k$. Note that by definition at the boundaries $P(\mathbf{n} + \mathbf{e}_i, t)|_{n_i=N} = 0$ and $P(\mathbf{n} - \mathbf{e}_i, t)|_{n_i=0} = 0$, so their summation indices go up to $n_i = N-1$, or start from $n_i = 1$, respectively.

After appropriate transformations of variables to only deal with $P(\mathbf{n},t)$ and re-indexing, we obtain

$$
\begin{aligned}
\frac{d\langle n_k \rangle}{dt} = \quad &\ldots \sum_{n_k=0}^{N} \ldots (n_k - 1) P(\mathbf{n}, t) T(\mathbf{n} \rightarrow \mathbf{n} - \mathbf{e}_k) \\
&+ \ldots \sum_{n_k=0}^{N} \ldots (n_k + 1) P(\mathbf{n}, t) T(\mathbf{n} \rightarrow \mathbf{n} + \mathbf{e}_k) \\
&- \ldots \sum_{n_k=0}^{N} \ldots n_k P(\mathbf{n}, t) T(\mathbf{n} \rightarrow \mathbf{n} + \mathbf{e}_k) \\
&- \ldots \sum_{n_k=0}^{N} \ldots n_k P(\mathbf{n}, t) T(\mathbf{n} \rightarrow \mathbf{n} - \mathbf{e}_k) \\
&+ \ldots \sum_{n_k=0}^{N} \sum_{n_i=0}^{N} \ldots n_k \sum_{i \neq k} P(\mathbf{n}, t) T(\mathbf{n} \rightarrow \mathbf{n} - \mathbf{e}_i) \\
&+ \ldots \sum_{n_k=0}^{N} \sum_{n_i=0}^{N} \ldots n_k \sum_{i \neq k} P(\mathbf{n}, t) T(\mathbf{n} \rightarrow \mathbf{n} + \mathbf{e}_i) \\
&- \ldots \sum_{n_k=0}^{N} \sum_{n_i=0}^{N} \ldots n_k \sum_{i \neq k} P(\mathbf{n}, t) T(\mathbf{n} \rightarrow \mathbf{n} + \mathbf{e}_i) \\
&- \ldots \sum_{n_k=0}^{N} \sum_{n_i=0}^{N} \ldots n_k \sum_{i \neq k} P(\mathbf{n}, t) T(\mathbf{n} \rightarrow \mathbf{n} - \mathbf{e}_i).
\end{aligned}
\tag{12}
$$

Note that the last 4 terms reduce to zero, and that at the boundaries $T(\mathbf{n} \rightarrow \mathbf{n} - \mathbf{e}_k)|_{n_k=0} = 0$ and $T(\mathbf{n} \rightarrow \mathbf{n} + \mathbf{e}_k)|_{n_k=N} = 0$, which allows including $n_k = 0$ and $n_k = N$ in the summations. Simplifying, we find

$$
\frac{d\langle n_k \rangle}{dt} = \quad \ldots \sum_{n_k=0}^{N} \ldots P(\mathbf{n}, t)(T(\mathbf{n} \rightarrow \mathbf{n} + \mathbf{e}_k) - T(\mathbf{n} \rightarrow \mathbf{n} - \mathbf{e}_k)),
\tag{13}
$$

and substituting the transition rates $T(\mathbf{n}{\rightarrow}\mathbf{n}{+}\mathbf{e}_i)$ and $T(\mathbf{n}{\rightarrow}\mathbf{n}{-}\mathbf{e}_i)$ from Eqs (10) leads to

$$
\frac{d\langle n_k \rangle}{dt} = f_k \left( \langle n_k \rangle - \sum_j \frac{\langle n_k n_j \rangle}{N} \right) + m_k \left( 1 - \sum_j \frac{\langle n_j \rangle}{N} \right) - \frac{\phi_k}{N} \langle n_k \rangle.
\tag{14}
$$

For other moments and models similar derivations can be done.

For the second moment, we find

$$
\begin{aligned}
\frac{d\langle n_k^2 \rangle}{dt} = \quad &\ldots \sum_{n_k=0}^{N} \ldots (1 + 2n_k) P(\mathbf{n}, t) T(\mathbf{n} \rightarrow \mathbf{n} + \mathbf{e}_k) \\
&+ \ldots \sum_{n_k=0}^{N} \ldots (1 - 2n_k) P(\mathbf{n}, t) T(\mathbf{n} \rightarrow \mathbf{n} - \mathbf{e}_k),
\end{aligned}
\tag{15}
$$

which after substituting $T(\mathbf{n}\to\mathbf{n}+\mathbf{e}_i)$ and $T(\mathbf{n}\to\mathbf{n}-\mathbf{e}_i)$ from Eqs (10) reduces to

$$
\begin{aligned}
\frac{d\langle n_k^2 \rangle}{dt} = \quad & f_k\left(\langle n_k \rangle - \sum_j \frac{\langle n_k n_j \rangle}{N} + 2\left(\langle n_k^2 \rangle - \sum_j \frac{\langle n_k^2 n_j \rangle}{N}\right)\right) \\
& + m_k\left(1 - \sum_j \frac{\langle n_j \rangle}{N} + 2\left(\langle n_k \rangle - \sum_j \frac{\langle n_k n_j \rangle}{N}\right)\right) \\
& + \frac{\phi_k}{N}\left(\langle n_k \rangle - 2\langle n_k^2 \rangle\right).
\end{aligned}
\tag{16}
$$

For the co-moments, we find

$$
\begin{aligned}
\frac{d\langle n_k n_l \rangle}{dt} = \quad & \ldots \sum_{n_k=0}^{N}\sum_{n_l=0}^{N} \ldots n_l P(\mathbf{n},t) T(\mathbf{n}\to\mathbf{n}+\mathbf{e}_k) \\
& + \ldots \sum_{n_k=0}^{N}\sum_{n_l=0}^{N} \ldots n_k P(\mathbf{n},t) T(\mathbf{n}\to\mathbf{n}+\mathbf{e}_l) \\
& - \ldots \sum_{n_k=0}^{N}\sum_{n_l=0}^{N} \ldots n_l P(\mathbf{n},t) T(\mathbf{n}\to\mathbf{n}-\mathbf{e}_k) \\
& - \ldots \sum_{n_k=0}^{N}\sum_{n_l=0}^{N} \ldots n_k P(\mathbf{n},t) T(\mathbf{n}\to\mathbf{n}-\mathbf{e}_l),
\end{aligned}
\tag{17}
$$

which after substituting $T(\mathbf{n}\to\mathbf{n}+\mathbf{e}_i)$ and $T(\mathbf{n}\to\mathbf{n}-\mathbf{e}_i)$ from Eqs (10) leads to

$$
\begin{aligned}
\frac{d\langle n_k n_l \rangle}{dt} = \quad & (f_k + f_l)\left(\langle n_k n_l \rangle - \sum_j \frac{\langle n_k n_l n_j \rangle}{N}\right) \\
& + m_k\left(\langle n_l \rangle - \sum_j \frac{\langle n_l n_j \rangle}{N}\right) + m_l\left(\langle n_k \rangle - \sum_j \frac{\langle n_k n_j \rangle}{N}\right) \\
& - \frac{\phi_k + \phi_l}{N}\langle n_k n_l \rangle.
\end{aligned}
\tag{18}
$$

Because each equation depends on even higher moments, e.g., $d\langle n_k n_l \rangle/dt$ depends on $\langle n_k n_l n_j \rangle$, it is not possible to solve this system of equations without additional assumptions. However, one can find approximate expressions, where lower moments replace higher moments. For example, $\langle n_k^2 n_j \rangle$ and $\langle n_k n_l n_j \rangle$ are approximated as functions of the lower moments: $\langle n_k^2 \rangle$, $\langle n_k n_l \rangle$, and $\langle n_j \rangle$. This technique, called moment closure approximation, leads to a closed system of ODEs and we use it in our approach. Various approximations stemming from numerical observations, physical considerations, or heuristics have been used with great success [42], and are available in automated software tools (e.g., MomentClosure.jl [39]). Kuehn [42] makes a thorough review of this technique. Here, we illustrate the approximation of third-order moments as the product of one co-moment and one moment, $\langle n_k n_l n_j \rangle \approx \langle n_k n_l \rangle \langle n_j \rangle$. The key of our approximation lies on the covariance of a pair of microbes and a single microbe being close to zero, $\langle n_k n_l, n_j \rangle \approx 0$, with 3 possible ways to distribute $k$, $l$, and $j$. We used $\langle n_k n_l n_j \rangle \approx \langle n_k n_l \rangle \langle n_j \rangle$ and $\langle n_k^2 n_j \rangle \approx \langle n_k^2 \rangle \langle n_j \rangle$ all along. The validity of our approximation can be tested by checking the covariances $\langle n_k n_l, n_j \rangle$ of the experimental data set, choosing a different approximation otherwise [42].

## Lotka–Volterra

Now, for a model with intra- and inter-specific interactions, let us define the transition rates,

$$T(\mathbf{n} \to \mathbf{n} + \mathbf{e}_i) = n_i(f_i + \sum_j A_{i,j} n_j) \tag{19a}$$

$$T(\mathbf{n} \to \mathbf{n} - \mathbf{e}_i) = n_i \sum_j B_{i,j} n_j, \tag{19b}$$

where $A$ and $B$ are positively defined matrices containing the interactions, satisfying $A_{i,j} = 0$ if $B_{i,j} > 0$, and $B_{i,j} = 0$ if $A_{i,j} > 0$. Ecologically, while interactions in $A$ promote growth, those in $B$ lead to death. Note that interactions ($i,j$ and $j,i$) can be asymmetrical. Finally, $f_i$ is the intrinsic growth rate.

For the first moment, similarly to Eq (13), we have

$$\frac{d\langle n_k \rangle}{dt} = \quad \ldots \sum_{n_k=0}^{\infty} \ldots P(\mathbf{n}, t)(T(\mathbf{n} \to \mathbf{n} + \mathbf{e}_k) - T(\mathbf{n} \to \mathbf{n} - \mathbf{e}_k)), \tag{20}$$

which after substituting $T(\mathbf{n} \to \mathbf{n} + \mathbf{e}_i)$ and $T(\mathbf{n} \to \mathbf{n} - \mathbf{e}_i)$ from Eqs (19),

$$\frac{d\langle n_k \rangle}{dt} = f_k \langle n_k \rangle + \sum_j \left( A_{k,j} - B_{k,j} \right) \langle n_k n_j \rangle, \tag{21}$$

takes the form of the conventional, deterministic Lotka–Volterra equations for the abundance with growth rate $f_k$ and interaction matrix $A_{k,j} - B_{k,j}$.

For the second moment, similarly to Eq (15)

$$\frac{d\langle n_k^2 \rangle}{dt} = \quad \ldots \sum_{n_k=0}^{\infty} \ldots (1 + 2n_k) P(\mathbf{n}, t) T(\mathbf{n} \to \mathbf{n} + \mathbf{e}_k)$$

$$+ \ldots \sum_{n_k=0}^{\infty} \ldots (1 - 2n_k) P(\mathbf{n}, t) T(\mathbf{n} \to \mathbf{n} - \mathbf{e}_k), \tag{22}$$

which after substituting $T(\mathbf{n} \to \mathbf{n} + \mathbf{e}_i)$ and $T(\mathbf{n} \to \mathbf{n} - \mathbf{e}_i)$ from Eqs (19) leads to

$$\frac{d\langle n_k^2 \rangle}{dt} = f_k(\langle n_k \rangle + 2\langle n_k^2 \rangle)$$

$$+ \sum_j (A_{k,j} + B_{k,j}) \langle n_k n_j \rangle + 2 \sum_j (A_{k,j} - B_{k,j}) \langle n_k^2 n_j \rangle. \tag{23}$$

For the co-moments, similarly to Eq (17), we derive

$$\frac{d\langle n_k n_l \rangle}{dt} = \quad \ldots \sum_{n_k=0}^{\infty} \sum_{n_l=0}^{\infty} \ldots n_l P(\mathbf{n}, t) T(\mathbf{n} \to \mathbf{n} + \mathbf{e}_k)$$

$$+ \ldots \sum_{n_k=0}^{\infty} \sum_{n_l=0}^{\infty} \ldots n_k P(\mathbf{n}, t) T(\mathbf{n} \to \mathbf{n} + \mathbf{e}_l)$$

$$- \ldots \sum_{n_k=0}^{\infty} \sum_{n_l=0}^{\infty} \ldots n_l P(\mathbf{n}, t) T(\mathbf{n} \to \mathbf{n} - \mathbf{e}_k)$$

$$- \ldots \sum_{n_k=0}^{\infty} \sum_{n_l=0}^{\infty} \ldots n_k P(\mathbf{n}, t) T(\mathbf{n} \to \mathbf{n} - \mathbf{e}_l), \tag{24}$$

which after substituting $T(\mathbf{n}\rightarrow\mathbf{n}+\mathbf{e}_i)$ and $T(\mathbf{n}\rightarrow\mathbf{n}-\mathbf{e}_i)$ from Eqs (19) reduces to

$$\frac{d\langle n_k n_l\rangle}{dt} = (f_k + f_l)\langle n_k n_l\rangle + \sum_j \left(A_{k,j} - B_{k,j} + A_{l,j} - B_{l,j}\right)\langle n_k n_l n_j\rangle. \tag{25}$$

As previously, a moment closure approximation is required to solve the system of equations. We used $\langle n_k n_l n_j\rangle \approx \langle n_k n_l\rangle\langle n_j\rangle$ and $\langle n_k^2 n_j\rangle \approx \langle n_k^2\rangle\langle n_j\rangle$.

## From absolute to relative abundance

The former equations account for the change in absolute abundance. To focus on relative abundance data, we define the relative abundance as follows:

$$x_k = \frac{n_k}{\sum_j n_j}, \tag{26}$$

and

$$n_\Sigma = \sum_j n_j, \tag{27}$$

to serve as a scaling factor. Thus,

$$n_k = x_k n_\Sigma. \tag{28}$$

Let us find the transformation to relative abundances for the first moment. Using the definition of the covariance $\langle x_k, n_\Sigma\rangle = \langle x_k n_\Sigma\rangle - \langle x_k\rangle\langle n_\Sigma\rangle$, such that $\langle x_k n_\Sigma\rangle = \langle x_k\rangle\langle n_\Sigma\rangle + \langle x_k, n_\Sigma\rangle$, we have

$$\begin{aligned}
\frac{d\langle n_k\rangle}{dt} &= \frac{d\langle x_k n_\Sigma\rangle}{dt} \\
&= \langle n_\Sigma\rangle\frac{d\langle x_k\rangle}{dt} + \langle x_k\rangle\frac{d\langle n_\Sigma\rangle}{dt} + \frac{d\langle x_k, n_\Sigma\rangle}{dt}.
\end{aligned} \tag{29}$$

Rearranging, the transformation is given by

$$\frac{d\langle x_k\rangle}{dt} = \frac{1}{\langle n_\Sigma\rangle}\left(\frac{d\langle n_k\rangle}{dt} - \langle x_k\rangle\frac{d\langle n_\Sigma\rangle}{dt} - \frac{d\langle x_k, n_\Sigma\rangle}{dt}\right). \tag{30}$$

For second-order moments, we use that $\langle x_k x_l n_\Sigma^2\rangle = \langle x_k x_l\rangle\langle n_\Sigma^2\rangle + \langle x_k x_l, n_\Sigma^2\rangle$ and approximate $\langle n_\Sigma^2\rangle \approx \langle n_\Sigma\rangle^2$. Then, using the chain rule

$$\begin{aligned}
\frac{d\langle n_k n_l\rangle}{dt} &= \frac{d\langle x_k x_l n_\Sigma^2\rangle}{dt} \\
&= \langle n_\Sigma\rangle^2\frac{d\langle x_k x_l\rangle}{dt} + 2\langle x_k x_l\rangle\langle n_\Sigma\rangle\frac{d\langle n_\Sigma\rangle}{dt} + \frac{d\langle x_k x_l, n_\Sigma^2\rangle}{dt}.
\end{aligned} \tag{31}$$

Rearranging, the transformations are given by

$$\frac{d\langle x_k x_l\rangle}{dt} = \frac{1}{\langle n_\Sigma\rangle^2}\left(\frac{d\langle n_k n_l\rangle}{dt} - 2\langle x_k x_l\rangle\langle n_\Sigma\rangle\frac{d\langle n_\Sigma\rangle}{dt} - \frac{d\langle x_k x_l, n_\Sigma^2\rangle}{dt}\right), \tag{32}$$

and

$$\frac{d\langle x_k^2 \rangle}{dt} = \frac{1}{\langle n_\Sigma \rangle^2}\left(\frac{d\langle n_k^2 \rangle}{dt} - 2\langle x_k^2 \rangle\langle n_\Sigma \rangle \frac{d\langle n_\Sigma \rangle}{dt} - \frac{d\langle x_k^2, n_\Sigma^2 \rangle}{dt}\right), \tag{33}$$

where the differential equation for $\langle n_\Sigma \rangle$ is given by

$$\frac{d\langle n_\Sigma \rangle}{dt} = \frac{d\langle \sum_j n_j \rangle}{dt} = \sum_j \frac{d\langle n_j \rangle}{dt}. \tag{34}$$

A close look at the dynamics of the covariances shows their contribution is negligible in large populations. To see this, let us write

$$\begin{aligned}
\frac{d\langle x_k, n_\Sigma \rangle}{dt} = \ &\ldots\sum_{n_k=0}^{\infty}\ldots\left(\frac{n_k}{n_\Sigma} - \langle\frac{n_k}{n_\Sigma}\rangle\right)(n_\Sigma - \langle n_\Sigma \rangle)P(\mathbf{n}+\mathbf{e}_k, t)T(\mathbf{n}+\mathbf{e}_k \to \mathbf{n}) \\
&+\ldots\sum_{n_k=0}^{\infty}\ldots\left(\frac{n_k}{n_\Sigma} - \langle\frac{n_k}{n_\Sigma}\rangle\right)(n_\Sigma - \langle n_\Sigma \rangle)P(\mathbf{n}-\mathbf{e}_k, t)T(\mathbf{n}-\mathbf{e}_k \to \mathbf{n}) \\
&-\ldots\sum_{n_k=0}^{\infty}\ldots\left(\frac{n_k}{n_\Sigma} - \langle\frac{n_k}{n_\Sigma}\rangle\right)(n_\Sigma - \langle n_\Sigma \rangle)P(\mathbf{n}, t)T(\mathbf{n} \to \mathbf{n}+\mathbf{e}_k) \\
&-\ldots\sum_{n_k=0}^{\infty}\ldots\left(\frac{n_k}{n_\Sigma} - \langle\frac{n_k}{n_\Sigma}\rangle\right)(n_\Sigma - \langle n_\Sigma \rangle)P(\mathbf{n}, t)T(\mathbf{n} \to \mathbf{n}-\mathbf{e}_k) \\
&+\ldots\sum_{n_k=0}^{\infty}\sum_{n_i=0}^{\infty}\ldots\frac{n_k}{n_\Sigma}(n_\Sigma - \langle n_\Sigma \rangle)\sum_{i\neq k}P(\mathbf{n}+\mathbf{e}_i, t)T(\mathbf{n}+\mathbf{e}_i \to \mathbf{n}) \\
&-\ldots\sum_{n_k=0}^{\infty}\sum_{n_i=0}^{\infty}\ldots\langle\frac{n_k}{n_\Sigma}\rangle(n_\Sigma - \langle n_\Sigma \rangle)\sum_{i\neq k}P(\mathbf{n}+\mathbf{e}_i, t)T(\mathbf{n}+\mathbf{e}_i \to \mathbf{n}) \\
&+\ldots\sum_{n_k=0}^{\infty}\sum_{n_i=1}^{\infty}\ldots\frac{n_k}{n_\Sigma}(n_\Sigma - \langle n_\Sigma \rangle)\sum_{i\neq k}P(\mathbf{n}-\mathbf{e}_i, t)T(\mathbf{n}-\mathbf{e}_i \to \mathbf{n}) \\
&-\ldots\sum_{n_k=0}^{\infty}\sum_{n_i=1}^{\infty}\ldots\langle\frac{n_k}{n_\Sigma}\rangle(n_\Sigma - \langle n_\Sigma \rangle)\sum_{i\neq k}P(\mathbf{n}-\mathbf{e}_i, t)T(\mathbf{n}-\mathbf{e}_i \to \mathbf{n}) \\
&-\ldots\sum_{n_k=0}^{\infty}\sum_{n_i=0}^{\infty}\ldots\left(\frac{n_k}{n_\Sigma} - \langle\frac{n_k}{n_\Sigma}\rangle\right)(n_\Sigma - \langle n_\Sigma \rangle)\sum_{i\neq k}P(\mathbf{n}, t)T(\mathbf{n} \to \mathbf{n}+\mathbf{e}_i) \\
&-\ldots\sum_{n_k=0}^{\infty}\sum_{n_i=0}^{\infty}\ldots\left(\frac{n_k}{n_\Sigma} - \langle\frac{n_k}{n_\Sigma}\rangle\right)(n_\Sigma - \langle n_\Sigma \rangle)\sum_{i\neq k}P(\mathbf{n}, t)T(\mathbf{n} \to \mathbf{n}-\mathbf{e}_i),
\end{aligned} \tag{35}$$

after the appropriate transformations of variable to only deal with $P(\mathbf{n},t)$ and re-indexing, we

find

$$
\begin{aligned}
\frac{d\langle x_k, n_\Sigma \rangle}{dt} = \quad & \dots \sum_{n_k=0}^{\infty} \dots \left( \frac{n_k - 1}{n_\Sigma - 1} - \langle \frac{n_k}{n_\Sigma} \rangle \right)(n_\Sigma - 1 - \langle n_\Sigma \rangle)P(\mathbf{n}, t)T(\mathbf{n} \to \mathbf{n} - \mathbf{e}_k) \\
& + \dots \sum_{n_k=0}^{\infty} \dots \left( \frac{n_k + 1}{n_\Sigma + 1} - \langle \frac{n_k}{n_\Sigma} \rangle \right)(n_\Sigma + 1 - \langle n_\Sigma \rangle)P(\mathbf{n}, t)T(\mathbf{n} \to \mathbf{n} + \mathbf{e}_k) \\
& - \dots \sum_{n_k=0}^{N} \dots \left( \frac{n_k}{n_\Sigma} - \langle \frac{n_k}{n_\Sigma} \rangle \right)(n_\Sigma - \langle n_\Sigma \rangle)P(\mathbf{n}, t)T(\mathbf{n} \to \mathbf{n} + \mathbf{e}_k) \\
& - \dots \sum_{n_k=0}^{N} \dots \left( \frac{n_k}{n_\Sigma} - \langle \frac{n_k}{n_\Sigma} \rangle \right)(n_\Sigma - \langle n_\Sigma \rangle)P(\mathbf{n}, t)T(\mathbf{n} \to \mathbf{n} - \mathbf{e}_k) \\
& + \dots \sum_{n_k=0}^{\infty}\sum_{n_i=0}^{\infty} \dots \frac{n_k}{n_\Sigma - 1}(n_\Sigma - 1 - \langle n_\Sigma \rangle)\sum_{i \neq k}P(\mathbf{n}, t)T(\mathbf{n} \to \mathbf{n} - \mathbf{e}_i) \\
& - \dots \sum_{n_k=0}^{\infty}\sum_{n_i=0}^{\infty} \dots \langle \frac{n_k}{n_\Sigma} \rangle(n_\Sigma - 1 - \langle n_\Sigma \rangle)\sum_{i \neq k}P(\mathbf{n}, t)T(\mathbf{n} \to \mathbf{n} - \mathbf{e}_i) \\
& + \dots \sum_{n_k=0}^{\infty}\sum_{n_i=1}^{\infty} \dots \frac{n_k}{n_\Sigma + 1}(n_\Sigma + 1 - \langle n_\Sigma \rangle)\sum_{i \neq k}P(\mathbf{n}, t)T(\mathbf{n} \to \mathbf{n} + \mathbf{e}_i) \\
& - \dots \sum_{n_k=0}^{\infty}\sum_{n_i=1}^{\infty} \dots \langle \frac{n_k}{n_\Sigma} \rangle(n_\Sigma + 1 - \langle n_\Sigma \rangle)\sum_{i \neq k}P(\mathbf{n}, t)T(\mathbf{n} \to \mathbf{n} + \mathbf{e}_i) \\
& - \dots \sum_{n_k=0}^{\infty}\sum_{n_i=0}^{\infty} \dots \left( \frac{n_k}{n_\Sigma} - \langle \frac{n_k}{n_\Sigma} \rangle \right)(n_\Sigma - \langle n_\Sigma \rangle)\sum_{i \neq k}P(\mathbf{n}, t)T(\mathbf{n} \to \mathbf{n} + \mathbf{e}_i) \\
& - \dots \sum_{n_k=0}^{\infty}\sum_{n_i=0}^{\infty} \dots \left( \frac{n_k}{n_\Sigma} - \langle \frac{n_k}{n_\Sigma} \rangle \right)(n_\Sigma - \langle n_\Sigma \rangle)\sum_{i \neq k}P(\mathbf{n}, t)T(\mathbf{n} \to \mathbf{n} - \mathbf{e}_i).
\end{aligned}
\tag{36}
$$

Note that if $n_\Sigma \gg 1$, $n_\Sigma \pm 1 \approx n_\Sigma$, then, if either $n_k \gg 1$, such that $n_k \pm 1 \approx n_k$ or $\frac{n_\Sigma - \langle n_\Sigma \rangle}{n_\Sigma} \approx 0$, the terms from the previous equation simplify, leading to

$$
\frac{d\langle x_k, n_\Sigma \rangle}{dt} \approx 0.
\tag{37}
$$

Similar arguments lead to conclude that,

$$
\frac{d\langle x_k x_l, n_\Sigma^2 \rangle}{dt} \approx 0,
\tag{38}
$$

and

$$
\frac{d\langle x_k^2, n_\Sigma^2 \rangle}{dt} \approx 0.
\tag{39}
$$

These approximations of the covariances are sensible in microbiomes, where $n_\Sigma, n_k \gg 1$ is often the case. Moreover, in the infinite population limit, covariances must be zero.

Putting all together, the approximated change of the first moment of relative abundance in large populations is given by

$$
\begin{aligned}
\frac{d\langle x_k\rangle}{dt} &= \frac{1}{\langle n_\Sigma\rangle}\left(\frac{d\langle n_k\rangle}{dt} - \langle x_k\rangle\sum_j\frac{d\langle n_j\rangle}{dt}\right)\\
&= \frac{1}{\langle n_\Sigma\rangle}\left(\frac{d\langle n_k\rangle}{dt} - \langle x_k\rangle\frac{d\langle n_\Sigma\rangle}{dt}\right),
\end{aligned}
\tag{40}
$$

while for the second moments of relative abundance

$$
\begin{aligned}
\frac{d\langle x_k x_l\rangle}{dt} &= \frac{1}{\langle n_\Sigma\rangle^2}\left(\frac{d\langle n_k n_l\rangle}{dt} - 2\langle x_k x_l\rangle\sum_j\frac{d\langle n_j\rangle}{dt}\right)\\
&= \frac{1}{\langle n_\Sigma\rangle^2}\left(\frac{d\langle n_k n_l\rangle}{dt} - 2\langle x_k x_l\rangle\langle n_\Sigma\rangle\frac{d\langle n_\Sigma\rangle}{dt}\right),
\end{aligned}
\tag{41}
$$

and

$$
\begin{aligned}
\frac{d\langle x_k^2\rangle}{dt} &= \frac{1}{\langle n_\Sigma\rangle^2}\left(\frac{d\langle n_k^2\rangle}{dt} - 2\langle x_k^2\rangle\sum_j\frac{d\langle n_j\rangle}{dt}\right)\\
&= \frac{1}{\langle n_\Sigma\rangle^2}\left(\frac{d\langle n_k^2\rangle}{dt} - 2\langle x_k^2\rangle\langle n_\Sigma\rangle\frac{d\langle n_\Sigma\rangle}{dt}\right).
\end{aligned}
\tag{42}
$$

As Joseph and colleagues [30], we see that the second term of each equation serves as a "correction factor" due to the fact that relative abundances must add up to one at all times. Finally, to solve these equations in terms of relative abundances only, changes of variable such as $\langle n_i\rangle \approx \langle x_i\rangle\langle n_\Sigma\rangle$, $\langle n_i n_j\rangle \approx \langle x_i x_j\rangle\langle n_\Sigma\rangle^2$, etc., are needed all along. These approximations are valid in large populations, where the covariance between relative abundance terms and $n_\Sigma$ are comparatively much smaller than the product of their averages.

### True parameters in simulations

Tables 1 and 2 contain the parameters used to simulate data.

### Inference settings

Tables 3–5 contain the probability priors for the inference of simulated and experimental data. Table 6 contains the settings for the inference of all data.

**Table 2. Parameters in simulated Lotka–Volterra (Fig 2B).** The interaction parameters as well as the growth rates were only chosen for illustration, thus, time units are arbitrary. We used relatively small initial populations for simplicity. However, larger initial populations can be easily tested.

| Microbial type | Interaction $I_{i,j}$ | | | Growth $f_i$ | Initial population size |
|---|---|---|---|---|---|
| | $j = 1$ | $j = 2$ | $j = 3$ | | |
| $i = 1$ | $-4.8 \cdot 10^{-5}$ | $2.7 \cdot 10^{-5}$ | $-5 \cdot 10^{-5}$ | 0.9 | 700 |
| $i = 2$ | $-2.9 \cdot 10^{-5}$ | $-6.3 \cdot 10^{-5}$ | $3.3 \cdot 10^{-5}$ | 1.7 | 7,000 |
| $i = 3$ | $2 \cdot 10^{-5}$ | $-4.1 \cdot 10^{-5}$ | $-5.4 \cdot 10^{-5}$ | 1.3 | 12,000 |

**Table 3. Priors for simulated logistic growth with immigration and death (Fig 2A).** A combination of uninformative (uniform) and informative (normal) priors were used for illustration. These priors span a wide range of values to test the ability of the inference workflow to find the true parameters in simulations (Table 1). $\mathcal{U}(a, b)$ indicates a uniform distribution in the range from $a$ to $b$. $\mathcal{N}(a, b)$ indicates a normal distribution with mean $a$ and standard deviation $b$.

| Parameter | Units | Prior |
|---|---|---|
| Growth $f_i$ | time$^{-1}$ | $\mathcal{U}(0, 2)$ |
| Death $\phi_i$ | cells/time | $\mathcal{U}(0, 200)$ |
| Immigration $m_i$ | cells/time | $\mathcal{N}(3,000, 1,000)$ |
| Shared carrying capacity $N$ | cells | $\mathcal{N}(10^5, 10^4)$ |
| Initial scaling factor $\bar{n}_\Sigma(0)$ | cells | $\mathcal{U}(4,000, 6,000)$ |

**Table 4. Priors for simulated Lotka–Volterra (Fig 2B).** A combination of uninformative (uniform) and informative (normal) priors were used for illustration. These priors span a wide range of values to test the ability of the inference workflow to find the true parameters in simulations (Table 2). $\mathcal{U}(a, b)$ indicates a uniform distribution in the range from $a$ to $b$. $\mathcal{N}(a, b)$ indicates a normal distribution with mean $a$ and standard deviation $b$.

| Parameter | Units | Prior |
|---|---|---|
| Growth $f_i$ | time$^{-1}$ | $\mathcal{N}(1.5, 0.5)$ |
| Intra-specific $I_{i,i}$ | (cells $\cdot$ time)$^{-1}$ | $\mathcal{U}(-10^{-4}, 0)$ |
| Inter-specific $I_{i,j}$ | (cells $\cdot$ time)$^{-1}$ | $\mathcal{N}(0, 5 \cdot 10^{-5})$ |
| Initial scaling factor $\bar{n}_\Sigma(0)$ | cells | $\mathcal{U}(1.5 \cdot 10^4, 2.5 \cdot 10^4)$ |

**Table 5. Priors for logistic growth with immigration and death in empirical mouse data (Fig 5).** Available evidence and back-of-the-envelope calculations (marked by *) were used to propose wide priors. $\mathcal{U}(a, b)$ indicates a uniform distribution in the range from $a$ to $b$. $\mathcal{N}(a, b)$ indicates a normal distribution with mean $a$ and standard deviation $b$.

| Parameter | Units | Prior | Motivation |
|---|---|---|---|
| Growth $f_i$ | day$^{-1}$ | $\mathcal{U}(0.25, 72)$ | cell division from 4 days to 20 min [43] |
| Death $\phi_i$ | cells/day | $\mathcal{U}(0, 2 \cdot 10^6)$ | from 0 to $\approx$23 cells/second* |
| Immigration $m_i$ | cells/day | $\mathcal{U}(0, 2 \cdot 10^6)$ | from 0 to $\approx$23 cells/second* |
| Shared carrying capacity $N$ | cells | $\mathcal{U}(1.4 \cdot 10^7, 1.6 \cdot 10^7)$ | max. number of cells in data [28] |

**Table 6. Settings for ABC-SMC code.** These settings were chosen to decrease the computing time, but still robustly minimize the distance between data and model. We used tools from the Python package pyABC [34], mainly *ABCSMC*. The maximum number of generations, mismatch threshold ($\varepsilon$) minimum, and minimum $\varepsilon$ change between generations are all stopping criteria (marked by *). LSODA is a numerical solver capable of selectively adapting to the stiffness of a system of differential equations. *NA*: not applicable.

| Setting | Simulations (Fig 2) | | Mouse data (Fig 5) |
|---|---|---|---|
| | **Logistic** | **Lotka–Volterra** | **Logistic** |
| Numeric integration method (*numpy*) | LSODA | | LSODA |
| ABC-SMC (*pyABC*) | | | |
| Calibration samples (to get first $\varepsilon$) | 2,000 | | 400 |
| * Max. number of generations | 15 | | 10 |
| Accepted samples per generation | Adaptive Population Size (CV = 0.4) | | APS (CV = 0.25) |
| Samples in first generation | 750 | | 2,000 |
| Min. samples per generation | 500 | | 500 |

*(Continued)*

**Table 6.** (Continued)

| Setting | Simulations ([Fig 2]) | | Mouse data ([Fig 5]) |
|---|---|---|---|
| | Logistic | Lotka–Volterra | Logistic |
| Max. samples per generation | 1,000 | | 1,000 |
| Mismatch threshold ($\varepsilon$) update | Quantile Epsilon (best 10%) | | QE (10%) |
| Strategy to sample parameters | Multivariate Normal Transition | | MNT |
| * Mismatch threshold ($\varepsilon$) min. | 0.046 | 0.071 | NA |
| * Min. $\varepsilon$ change btw. generations | NA | | $10^{12}$ |

## Supporting information

**S1 Fig. Effect of 2 alternative distance metrics on the inference outcome of the Lotka–Volterra model.** We inferred all parameters from relative abundance simulated data as shown in Fig 2. However, for the distance metric between model and data, Eq (4), statistical moments were rescaled or not. While for absolute abundance, the second-order moments and co-moments are naturally larger than the first-order moments, for relative abundance data, the opposite is true. Rescaling the moments can modify their importance during the inference process. To test this, we took the square root (of the squared errors) of second-order moments and co-moments for absolute abundance, and of first-order moments for relative abundance data. The posteriors of rescaled and non-rescaled moments largely overlap, with non-rescaled moments (our approach in the other figures) leading to more certainty. The data underlying this figure can be found in https://doi.org/10.5281/zenodo.13958305.
(TIF)

**S2 Fig. Effect of data measurement noise on the uncertainty of inferred Lotka–Volterra parameters.** We inferred all parameters from simulated data as shown in Fig 2. To show the effect of noise on all parameters, we computed the L-2 norm of relative errors of the parameters (Table 2). We simplified the nuances of empirical noise assuming a scenario where all microbial abundances are affected proportionally. Concretely, a uniform noise distribution was shared among all microbial types and constant through time. For low noise, data could be altered by up to ±5%, while for medium and high noise, by up to ±10% and ±20%. Noise was sampled independently for each microbial type at each time point, affecting their absolute abundance from which relative abundances were computed. The data underlying this figure can be found in https://doi.org/10.5281/zenodo.13958305.
(TIF)

## Acknowledgments

We thank the Theoretical Biology Department in the MPI Plön, the Collaborative Research Centre 1182: Origin and Function of Metaorganisms and Wenying Shou for fruitful discussions.

## Author Contributions

**Conceptualization:** Román Zapién-Campos, Arne Traulsen.

**Formal analysis:** Román Zapién-Campos, Florence Bansept.

**Investigation:** Román Zapién-Campos.

**Methodology:** Román Zapién-Campos, Florence Bansept, Arne Traulsen.

**Project administration:** Arne Traulsen.

**Software:** Román Zapién-Campos.

**Supervision:** Arne Traulsen.

**Validation:** Florence Bansept.

**Visualization:** Román Zapién-Campos, Florence Bansept, Arne Traulsen.

**Writing – original draft:** Román Zapién-Campos.

**Writing – review & editing:** Román Zapién-Campos, Florence Bansept, Arne Traulsen.

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
