## [Editor Report · Decision Letter 0]

27 Feb 2024

Dear Dr Zapién-Campos, 

Thank you for submitting your manuscript entitled "Inferring interactions from microbiome data" for consideration as a Research Article by PLOS Biology.

Your manuscript has now been evaluated by the PLOS Biology editorial staff, as well as by an academic editor with relevant expertise, and I am writing to let you know that we would like to send your submission out for external peer review.

IMPORTANT: We would like to review your paper as a Methods and Resources paper. There's no need for any re-formatting, but please select "Methods and Resources" as the article type when you upload the additional metadata (see next paragraph)

Once your full submission is complete, your paper will undergo a series of checks in preparation for peer review. After your manuscript has passed the checks it will be sent out for review. To provide the metadata for your submission, please Login to Editorial Manager (https://www.editorialmanager.com/pbiology) within two working days, i.e. by Feb 29 2024 11:59PM.

Kind regards,

Melissa

Melissa Vazquez Hernandez, Ph.D.

Associate Editor

PLOS Biology

---

## [Decision Letter · Decision Letter 1]

9 Apr 2024

Dear Dr Zapién-Campos,

Thank you for your patience while your manuscript "Inferring interactions from microbiome data" was peer-reviewed at PLOS Biology. It has now been evaluated by the PLOS Biology editors, an Academic Editor with relevant expertise, and by three independent reviewers. 

In light of the reviews, which you will find at the end of this email, we would like to invite you to revise the work to thoroughly address the reviewers' reports. As you will see below, majority of reviewers are positive about the relevance of the model, yet some concerns have been raised during revision. Reviewer #1 has multiple questions about the model, and suggests some additions to the model that could help in the computational feasibility of the approach. In addition, reviewer #2 suggests to increase the number of in silico systems tested, to test the method on time series at steady state, and to compare the model with other methods. Reviewer #3 asks to test the model with temporal networks and multi-layer networks, and to discuss how other forms of interactions could affect the stochastic model. Of particular importance are the points raised related to providing clear instructions to experimentalists. We agree with all reviewer concerns and they should be addressed to further consider work for publication in PLOS Biology.

Given the extent of revision needed, we cannot make a decision about publication until we have seen the revised manuscript and your response to the reviewers' comments. Your revised manuscript is likely to be sent for further evaluation by all or a subset of the reviewers.

**IMPORTANT - SUBMITTING YOUR REVISION**

*Re-submission Checklist*

*Published Peer Review*

*PLOS Data Policy*

*Blot and Gel Data Policy*

Sincerely,

Melissa

Melissa Vazquez Hernandez, Ph.D.

Associate Editor

PLOS Biology

REVIEWERS' COMMENTS:

Reviewer #1: Please see attached file.

Reviewer #2: 

The manuscript is very well written. The problem tackled is important and the approach is to my knowledge new for microbial time series. 

The authors propose to use the equations for the first moments of the distribution of microbial abundances in replicate communities, those equations are derived from a master equation of the logistic model or the generalized Lotka-Volterra model. Parameter fitting is performed with a method called Approximate Bayesian Computation - Sequential Monte Carlo. 

Major comments: 

- Only three different in silico systems have been tested. I would have appreciated a more extensive study to determine under which conditions model parameters can be inferred by the method. One would eventually want to be able to give instructions to experimentalists to tell for instance how densely sampled the system should be and how many replicates we need to be able to rely on this method. 

- Additionally, it is important to test the method on time series which are at steady state. It is often the case that for in vivo time series, we only have access to fluctuations around the steady state. 

- Comparison with other method could also be done. Could we rely on the method of Stein et al, 2013 with one time series (instead of averaging) and get the same parameters? It is said that one advantage of the proposed workflow is that "the dynamics of the moments uses more information contained in the data", but it is not needed to average the data to apply the method of Stein et al. for instance (or do I miss something?). If replicates are available, we could apply Stein's method multiple times and then compute the averages of the parameters and their variabilities. If we do so, would we get the same result as with your method? 

- It seems that the method can only be applied if the initial conditions are identical in the replicates. This seems hard to realize for in vivo microbial communities. Are they ways to circumvent this limitation? 

- [this question is similar to the previous one] It is not clear to me which type of experimental data can be aggregated and modeled as P(n,t). Can they be coming from different environmental conditions, different initial conditions, different species compositions? Or do we need very well controlled experimental conditions? 

- Line 284: I would want to know if in typically experimental microbial time series the approximation <n_k n_i n_j> \\sim <n_k n_k> <n_j> is true. Is this easy to check? 

- For the mice experimental data, why didn't you try fitting a generalized Lotka-Volterra model? From the conclusions of Grilli, Nature Communications 2020 and Descheemaeker eLife 2020, this makes sense as the time series analyzed in those papers concluded that the characteristics of the fluctuations around steady states can be explained with a stochastic logistic model with linear noise, and therefore no pairwise interactions (however transient dynamics is very limited in the time series analyzed). 

Minor comments/questions: 

- Line 141: I do not know what identifiability tableaus means. 

- For equation (10a), I guess N means the maximal number of microbes in the system. If that is correct, I would say it explicitly. 

- In equation (10b) it is not directly obvious to me why the death rate would decrease in a larger population (i.e. a population with a larger N). 

- For the simulated time series, you used N=10^5. Can you simulate realistic numbers? For human gut microbial communities, the number of microbes is much larger. 

Reviewer #3: 

The manuscript entitled "Inferring interactions from microbiome data", provides a significant contribution to the field of theoretical microbiology. The authors highlight the limitations of fitting deterministic models to averaged data, which often result in the loss of valuable information. To address these limitations, the manuscript proposes a stochastic model to infer distributions of interaction parameters that best describe biological experimental data.

Starting with a stochastic model, the authors derive dynamical equations for not only the average but also higher statistical moments of microbial abundances. These equations are utilized to infer distributions of interaction parameters, thereby improving identifiability and precision in describing the experimental data. Compared to existing approaches, this method offers a more comprehensive understanding of interactions in microbiomes.

Overall, I have really enjoyed reading this manuscript. The work presents a compelling and innovative approach to study microbial interactions within microbiomes. The problem of integrating theoretical models with experimental data in microbiome research is well-motivated, and the proposed method represents a significant advancement in addressing this challenge. I am in principle in favor of publication, but subject to the following major and minor suggestions for improvement.

Major suggestions

1. While the manuscript significantly demonstrates the effectiveness of combining Bayesian inference and stochastic modeling to infer interactions from microbiome data, it could benefit from a clearer explanation of how this approach surpasses classical deterministic methods. Additionally, it would also be more informative to compare and contrast the advantages and disadvantages of the proposed model with existing models in the field.

2. The impact of underlying network structures on microbial community dynamics is well-recognized. It would be beneficial to explore the applicability of this method to a range of network structures, including temporal networks (Li et al., Science 358: 1042-1046, 2017; Nat. Commun. 11: 2259, 2020) and multi-layer networks (Wang et al., J. R. Soc. Interface 20: 20220752, 2023). Such an analysis could provide a more comprehensive understanding of the method's potential and limitations in various network contexts within microbial communities.

3. In addition to pairwise interactions, it is important to consider other forms of interactions in this stochastic model, such as higher-order interactions and time delay interactions (Yang et al. Nat. Ecol. Evol. 7: 1610-1619, 2023). Discussing these interaction types would enrich the model and provide a more comprehensive understanding of microbial community dynamics. It is also meaningful to discuss the effects of this stochastic model on different ecological traits, such as stability and reactivity. Exploring these implications would provide valuable insights into the broader ecological implications of this stochastic model.

Minor suggestions

1. The title appears somewhat confusing. A more detailed and descriptive title would enhance the clarity of this manuscript.

2. Discussions on the limitations of the stochastic model are not sufficiently thorough and could be expanded to provide a more comprehensive assessment.

3. The figure captions should include specific parameter values to enhance the clarity and reproducibility of the results.

4. A more detailed discussion is needed regarding the application of this model, including its practical implications in real-world scenarios.

Apart from this, I am happy to congratulate the authors for their inspiring work and reiterate my recommendation for acceptance in PLoS Biology.

---

## [Decision Letter · Decision Letter 2]

29 Sep 2024

Dear Dr Zapién-Campos,

Thank you for your patience while we considered your revised manuscript "Inferring microbiome interactions from distributions of time series replicates" for publication as a Methods and Resources at PLOS Biology. This revised version of your manuscript has been evaluated by the PLOS Biology editors, the Academic Editor and the original reviewers. I am truly sorry for the long process this has taken us. 

Based on the reviews and on our Academic Editor's assessment of your revision, we are likely to accept this manuscript for publication, provided you satisfactorily address the remaining points raised by the reviewers. Please also make sure to address the following data and other policy-related requests.

a) We routinely suggest changes to titles to ensure maximum accessibility for a broad, non-specialist readership, and to ensure they reflect the contents of the paper. In this case, we would suggest a minor edit to the title, as follows. Please ensure you change both the manuscript file and the online submission system, as they need to match for final acceptance:

"Stochastic models allow improved inference of microbiome interactions from time series data"

b) Thank you for indicating the agencies that have funded the project. Please also provide the grant number.

Please supply the numerical values either in the a supplementary file or as a permanent DOI’d deposition for the following figures:

Figure 2ABD, 3, 4AB, 5ABC, S1, S2AB

d) Please cite the location of the data clearly in all relevant main and supplementary Figure legends, e.g. “The data underlying this Figure can be found in S1 Data” or “The data underlying this Figure can be found in https://doi.org/10.5281/zenodo.XXXXX”

e) Thank you for providing the data in the Zenodo Repository. However, I will need you to provide the DOI so we can get access to it before acceptance.

f) Please ensure that your Data Statement in the submission system accurately describes where your data can be found and is in final format, as it will be published as written there.

g) Per journal policy, if you have generated any custom code during the course of this investigation, please make it available without restrictions upon publication. Please ensure that the code is sufficiently well documented and reusable, and that your Data Statement in the Editorial Manager submission system accurately describes where your code can be found.

We expect to receive your revised manuscript within two weeks. 

*Published Peer Review History*

*Press*

Sincerely,

Melissa

Melissa Vazquez Hernandez, Ph.D.

Associate Editor

PLOS Biology

REVIEWERS' COMMENTS:

Reviewer #1: 

The authors have largely addressed my original comments, however I have two minor points that remain.

1. The authors claim that the method is not limited by the number of states. I can't see how this is true. If the authors have, say, 20 states, with covariances and means (first and second order moments), the authors would have to analyse a system with, effectively, 200+ states. Can the method really handle this?

2. The authors claim that they cannot use a likelihood based method unless the SDE admits an analytical solution. However, I would expect the moment-matching ABC approach to perform similarly (but much more slowly) to an approximate likelihood-based approach where a Gaussian likelihood with a matched mean and covariance is used.

Reviewer #2: 

Overall, I believe the manuscript has improved, and most of the comments have been addressed. However, I still think there are some areas that require further refinement.

Regarding the recommendations for experimentalists, would you suggest that 4 replicates and 3 time points are sufficient? I would assume it is also important to specify the intervals between time points, as well as emphasize that the time points should be captured during the transient dynamics phase. From Fig. 4, you conclude that it is more beneficial to increase the number of time points than to increase the number of replicates. However, in the extreme case where additional time points are only taken at steady-state, I suspect it might be more advantageous to increase the number of replicates. I believe the conclusion drawn from Fig. 4 could be more nuanced.

You mentioned that the method could be applied to snapshots. Could you provide an example where a single time point is sufficient to recover the parameters? Or did you mean that while the method could be applied to snapshots, it would not be effective?

Is there any way to estimate the noise level in the OMM dataset, for instance? If so, this would strengthen Fig. 4, as you could argue that the noise level in the experimental dataset is comparable to the one for this dataset.

Regarding Fig. 5C, I am uncertain whether measuring uncertainty as s.d.post/s.d.prior is the best approach for assessing the quality of the inference. I would prefer reporting the posterior standard deviation alone. If the prior has a large width, the ratio might decrease, but the actual uncertainty in the estimation would not. Or am I misunderstanding something?

Reviewer #3: 

The authors have made great efforts in addressing my comments. I am happy to recommend the acceptance, but please check there are many question marks on the current version when preparing the final version.

---

## [Editor Report · Decision Letter 3]

24 Oct 2024

Dear Román,

Thank you for the submission of your revised Methods and Resources "Stochastic models allow improved inference of microbiome interactions from time series data" for publication in PLOS Biology. On behalf of my colleagues and the Academic Editor, Isabel Gordo, I am pleased to say that we can in principle accept your manuscript for publication, provided you address any remaining formatting and reporting issues. These will be detailed in an email you should receive within 2-3 business days from our colleagues in the journal operations team; no action is required from you until then. Please note that we will not be able to formally accept your manuscript and schedule it for publication until you have completed any requested changes.

PRESS

Sincerely, 

Melissa

Melissa Vazquez Hernandez, Ph.D., Ph.D.

Associate Editor

PLOS Biology
